# Zinc(II)—The Overlooked Éminence Grise of Chloroquine’s Fight against COVID-19?

**DOI:** 10.3390/ph13090228

**Published:** 2020-09-01

**Authors:** Aleksandra Hecel, Małgorzata Ostrowska, Kamila Stokowa-Sołtys, Joanna Wątły, Dorota Dudek, Adriana Miller, Sławomir Potocki, Agnieszka Matera-Witkiewicz, Alicia Dominguez-Martin, Henryk Kozłowski, Magdalena Rowińska-Żyrek

**Affiliations:** 1Faculty of Chemistry, University of Wroclaw, F. Joliot-Curie 14, 50-383 Wroclaw, Poland; aleksandra.hecel@chem.uni.wroc.pl (A.H.); malgorzata.ostrowska@chem.uni.wroc.pl (M.O.); kamila.stokowa-soltys@chem.uni.wroc.pl (K.S.-S.); joanna.watly@chem.uni.wroc.pl (J.W.); dorota.dudek@chem.uni.wroc.pl (D.D.); adriana.miller@chem.uni.wroc.pl (A.M.); slawomir.potocki@chem.uni.wroc.pl (S.P.); henryk.kozlowski@chem.uni.wroc.pl (H.K.); 2Screening Laboratory of Biological Activity Tests and Collection of Biological Material, Faculty of Pharmacy, Wroclaw Medical University, Borowska 211A, 50-556 Wroclaw, Poland; agnieszka.matera-witkiewicz@umed.wroc.pl; 3Department of Inorganic Chemistry, Faculty of Pharmacy, University of Granada, E-18071 Granada, Spain; adominguez@ugr.es; 4Department of Physiotherapy, Opole Medical School, Katowicka 68, 40-060 Opole, Poland

**Keywords:** COVID-19, SARS-CoV-2, 2019-nCoV, Zn(II) ionophores, chloroquine, hydroxy- chloroquine

## Abstract

Zn(II) is an inhibitor of *SARS-CoV-2*′s RNA-dependent RNA polymerase, and chloroquine and hydroxychloroquine are Zn(II) ionophores–this statement gives a curious mind a lot to think about. We show results of the first clinical trials on chloroquine (CQ) and hydroxychloroquine (HCQ) in the treatment of COVID-19, as well as earlier reports on the anticoronaviral properties of these two compounds and of Zn(II) itself. Other FDA-approved Zn(II) ionophores are given a decent amount of attention and are thought of as possible COVID-19 therapeutics.

## 1. Introduction

Scientific reports usually begin with a brief rationale which explains why the given topic was undertaken—studied/reviewed/thought of. In the case of this work, we strongly feel that in July 2020, COVID-19, the cause of severe respiratory failure and over 673,169 deaths worldwide, and at the same time, the origin of an upcoming global economic crisis, does not need to be introduced to anyone. To the best of our knowledge, no specific and effective pharmacological treatment has yet been established, although more than 2834 clinical studies have already been registered (as of 30 July 2020).

The core idea of this work was initially based on a link between two papers—one, which states that Zn(II) is a potent inhibitor of coronavirus RNA-based polymerase [1] and another, which shows that chloroquine, a common antimalarial drug, also used against several autoimmune diseases, is a zinc ionophore [2]. This core idea is sustained by: (i) the results of the first independent clinical trials which show the clinical benefit of chloroquine (CQ) and hydroxychloroquine (HCQ) treatment in COVID-19 infected patients [3,4], and further built on: (ii) numerous reports on the anticoronaviral and antiviral properties of CQ and HCQ, (iii) antiviral properties of Zn(II) and Zn(II) ionophores, and (iv) ongoing clinical trials which include Zn(II) in CQ/HCQ treatments.

Could the mode of action, or at least one of the possible modes of action of CQ/HCQ be as simple as being an ionophore, thus bringing more Zn(II), a potent RNA-based polymerase inhibitor, into the cell? At this place, it would be honest to add, that this question remains a rhetorical one also at the end of this paper—clinical studies that aim to answer it are still ongoing. Zinc(II) could turn out to be the éminence grise, but it could also turn out to be one of the pieces of a really big puzzle.

We do not provide the reader with a definite answer, but elaborate more on this idea, shedding light on Zn(II) as a possible RNA-dependent RNA polymerase inhibitor and on the possible anticoronaviral and antiviral mode of action of zinc(II) ionophores. We show the first success and the (impressive) amount of interest in chloroquine and hydroxychloroquine in COVID-19 clinical trials and also briefly summarize the massive amount of ongoing COVID-19 clinical trials on non-Zn(II) ionophore treatments. It is important to add that the latter task is not a trivial one, since the amount of COVID-19 clinical trials increases at an incredible rate; while many others are already in progress, not yet registered, but already approved by the corresponding bioethics committees.

The situation with the use of HQ itself as a COVID-19 drug is ‘very dynamic’, and even as this article was being written, several articles came out on the results of clinical trials, often ambiguous in their conclusions, ranging from extremely positive in a study involving 550 Chinese patients [5] to no benefit in a study on 181 French patients [6]. Such “dynamic” and “hot” topics are very likely to attract cases of scientific misconduct—this was the case of a famous Lancet paper by Mehra et al. [7], published on May 22nd, discussing a huge amount of medical data (over 96,000 patients from 671 hospitals) which, according to the authors, showed that chloroquine/hydroxychloroquine therapy was ineffective and, what is more distressing, increased the risk of ventricular arrhythmias and in-hospital deaths with COVID-19. After that, the situation becomes “dynamic”: based on this publication, on May 25th, the WHO decides to withdraw some of the clinical tests with HQ/HCQ [8]. On June 3rd, Lancet publishes an Expression of Concern [9], which mentions the presence of “serious scientific questions” and errors in the article, and an audit is commissioned by three of the four authors not connected to Surgisphere—the company responsible for the data collection and analysis. Two days later, the same three of the four authors of the mentioned article ask the editor to retract the paper. According to the authors’ announcement, after having been rejected access to the Surgisphere patients’ databases to the independent auditors, they could not vouch for the veracity of the data any longer [10]. At the same time, a similar situation happens to an New England Journal of Medicine paper on the safety concern of chloroquine [11], which relied on data from the same company which took part in the Lancet study.

At the time this article was being written, several excellent preprints on the combination of CQ/HCQ came out: the first one by Scholz and Derwand, who also suggest that CQ and HCQ can inhibit SARS-CoV-2 replication by increasing pH in intracellular vesicles and by targeting extracellular zinc(II) to lysosomes, where Zn(II) then acts as an RdRp inhibitor [12]. They also point out that zinc(II) deficiency often occurs in those patients, who are at higher risk of a severe course of COVID-19—those with cardiovascular disease, chronic pulmonary disease, or diabetes. Another work by Scholz et al. [13] describes outcomes of the treatment of COVID-19 patients after early treatment with zinc(II), low dose hydroxychloroquine, and azithromycin (the triple therapy) dependent on risk stratification. The study, which includes a treatment group of 141 COVID-19 patients (median age 58 years) treated with the triple therapy for 5 days, shows that the triple therapy treatment of COVID-19 should be included as early as possible after symptom onset. The zinc(II) and hydroxychloroquine treated group was associated with significantly less hospitalizations (84% less than in the untreated control group) and 5 times less all-cause deaths (one patient (0.7%) died in the treatment group versus 13 patients (3.5%) in the untreated group). The study further underlines the importance of the early onset of Zn(II) treatment, which makes perfect sense if we consider Zn(II) as an RNA-dependent RNA polymerase inhibitor (as discussed in Section 2).

Another preprint shows the first in vivo evidence that zinc sulfate in combination with hydroxychloroquine may play a role in therapeutic management for COVID-19—administration of zinc sulfate together with HQ to 411 patients resulted in decreased mortality, admission to the ICU, and the need for ventilation [14].

This additionally encouraged us to look for more FDA-approved Zn(II) ionophores (or even those which are present in food) with antiviral, but also anticancer and antibacterial properties—because, if the mode of action of chloroquine and hydroxychloroquine really is based on their Zn(II) ionophoric activity, why not focus on much safer, clinically used, or naturally occurring zinc(II) ionophores with less side effects? In Section 5, Section 6 and Section 7, we summarize the current knowledge on FDA–approved Zn(II) ionophores with antiviral, anticancer and/or antibacterial activity. Could they be of any use in the anti-COVID fight?

## 2. Zn(II) as an RNA-Dependent RNA Polymerase Inhibitor

RNA-dependent RNA polymerase (RdRp) is the core enzyme in the replicative cycle of all positive-strand RNA (+RNA) viruses, which catalyzes the replication of RNA from a RNA template in a two-step mechanism [15,16,17]: (i) the initiation step of RNA synthesis begins at the 3′ end of the RNA template and the (ii) elongation phase, in which the nucleotidyl transfer is repeated with subsequent NTPs to generate the complementary RNA product.

Coronaviruses (order *Nidovirales*) are a family of +RNA viruses with the largest single-stranded RNA genomes (~30kB) known to date [18,19,20]. The unusual complex replication and transcription machinery of severe acute respiratory syndrome coronavirus (SARS-CoV) is composed of 16 nonstructural proteins (nsps), produced during co- and posttranslational proteolytic processing of two replicase polyproteins (pp1a and pp1ab). The canonical RdRp is localized in the C-terminal part of nonstructural protein 12 (nsp12) and possesses an architecture common to all viral polymerases. Their activity is greatly stimulated by the nsp7 and nsp8 co-factors, and CoV RdRp which belongs to the class of primer-dependent polymerases. The overall architecture of the CoV nsp12-nsp7-nsp8 complex is well known and widely described in literature, and it was recently shown that SARS-CoV-2nsp12-nsp7-nsp8 complex has a conserved structure typical for SARS-CoV viruses (Figure 1a). The nsp7-nsp8 pair of SARS-CoV-2 shows a structure, similar to the SARS-CoV nsp7-nsp8 pair, with 13 additional amino acid residues only [21,22]. The nsp12 polymerase binds to the nsp7-nsp8 heterodimer with the second subunit of nsp8 occupying a distinct binding site. SARS-CoV-2 nsp12 contains a polymerase domain (a.a. 367–920) that adopts a cupped “right hand” like structure similar to other polymerases. The polymerase is composed of three subdomains: the fingers domain (a.a. 366–581 and 621–679), a palm domain (a.a. 582–620 and 680–815) and a thumb domain (a.a. 816–920) (Figure 1a). SARS-CoV-2 nsp12 also contains a nidovirus-unique N-terminal extension domain (a.a. 60–249) that adopts a nidovirus RdRp-associated nucleotidyltransferase (NiRAN) architecture. These two domains are connected by an interface domain (a.a 250–365) (Figure 1a). The active site of SARS-CoV-2 RdRp has typically seven conserved motif regions (A–G) involved in template and nucleotide binding and catalysis. The mechanism by which SARS-CoV-2 polymerases carry out de novo RNA initiation is similar to those described for SARS-CoV and for other RNA polymerases such as HCV and PV polymerase [22]. The incoming nucleoside triphosphate (NTP) enters the active site through a channel formed by a set of hydrophilic residues in motif F. The RNA template is expected to enter the active site composed of motifs A and C through a groove clamped by motif F and G. Motif E and the thumb subdomain support the primer strand. The product-template hybrid exits the active site through the RNA exit tunnel at the front side of the polymerase [21,22].

Taking into account the essential role of the RdRp in the virus life cycle, as well as the effectiveness of polymerase inhibitors used in the treatment of other viral infections (HIV-1, HBV, HCV and herpes virus), the SARS-CoV-2 RdRp is widely recognized as one of the main, important and thus attractive targets for the rational design of anti-CoV drugs [23,24]. Zn(II) is naturally present in the structure of 2019-nCOV’s RdRp (Figure 1), and its elevated amounts clearly inhibit the course of infection. There is a growing amount of evidence that one of the promising candidates to have direct inhibitory effect on the replicative cycle of SARS-CoV-2 might be Zn(II).

Zn(II) is a generally known immune system enhancer as well as viral inhibitor—numerous studies have shown that intracellular Zn(II) levels have an effect on the life cycle of various RNA viruses that infect the human respiratory tract, including influenza [25], respiratory syncytial virus [26] and several picornaviruses [27,28,29,30,31,32]. Zinc(II) also inhibits the replication of DNA-containing viruses: human immunodeficiency virus-1 (HIV-1) [33], vaccinia virus [34], and polioviruses [35] in vitro. Several potential mechanisms have been proposed for the observed zinc(II)-mediated inhibition of RNA viruses replication in vitro [29,36,37]; for some RNA viruses, the disruption of replication has been attributed to the interference with viral protein processing [32], while for other, the direct inhibition of RdRp activity has been suggested [1,38,39,40].

Studies on the effectiveness of heavy metal chelating agents for their inhibitory effect on the influenza RdRp, reveal both surprising and promising results. Now, one could expect that chelating Zn(II) could, at least theoretically, inhibit the proper folding of RdRp; instead, the contrary was observed: when Zn(II) ions were added to the influenza RNA polymerase enzyme complex, they showed inhibitory activity, but the inhibition effect of the addition of Zn(II)-chelator complexes was greater than the effect of the chelating agent itself.

It was suggested that Zn(II)-chelator complexes dissociate in solution, after which Zn(II) is absorbed by proteins, including polymerase polypeptide(s), and thus exert a separate inhibitory effect on RdRp [25,38]. An inhibitory effect of Zn(II) ions with an IC_50_ (inhibitory concentration: median value of inhibitor, where 50% of biological and biochemical function of organism inhibition is observed) of about 60 mM on HCV RdRp activity was demonstrated. The authors postulated two possible mechanisms of the observed scenario: (i) Zn(II) ions compete with Mg(II) and Mn(II) ions for binding to the carboxylate of three aspartic acids from A and C motifs, or (ii) zinc(II) ions bind to another site, different from the catalytic one, and form an allosteric obstacle caused directly by a nucleophilic attack on the α-phosphate of the 3′-hydroxyl group of the primer [39]. Another work by Hung et al. showed that Zn(II) inhibits HRV-16 3D polymerase with an IC_50_ as low as 0.6 mM, but the authors did not believe in a possible displacement of the catalytically essential Mg(II) ion due to the significantly lower concentration of Zn(II) compared to Mg(II) [40]. At this point it is important to underline that the discussed mechanisms are still hypotheses, and the exact mechanism has not yet been proved in detail.

The inhibitory effect of Zn(II) ions on RdRp was also observed for two major pathogens of humans and livestock, severe acute respiratory syndrome coronavirus and the arteriviruses equine arteritis virus (EAV), respectively [1]. Zn(II) concentrations as low as 2 µM appear to be a specific inhibitor of the initiation and elongation phases of RNA synthesis. The authors suggested that Zn(II) does not compete with Mg(II) and binds to another Zn(II)-specific binding site [1]. In 2019 Kirchdoerfer et al. confirmed, for SARS-CoV, two metal binding-sites to which they have assigned Zn(II) binding. These binding sites are highly conserved across the CoV family [21].

As previously predicted for the SARS-CoV, also SARS-CoV*-2* nsp12 contains two metal-binding sites to which Zn(II) ions are bound. The Zn(II) ions in both viruses are present in the same location: the first binding site is in the interface domain (His295, Cys301, Cys306 and Cys310) and the second is in the fingers domain (Cys487, His642, Cys645 and Cys646) (Figure 1b and c, respectively). Both of these metal-binding sites are away from known active sites as well as protein–protein and protein–RNA interactions. Therefore, it is thought that Zn(II) ions play a structural role in the folded protein rather than are directly involved in enzymatic activity [21,22]. An in-depth structural analysis and subsequent mutational studies targeting aforementioned metal-binding sites are required to provide further insight into and a structural basis for the Zn(II)-induced inhibitory effects on RdRp activity. What is clear so far is the fact that the combination of Zn(II) ions and Zn(II)-ionophores inhibits the nidovirus replication in cell culture. If the Zn(II) ions can inhibit coronavirus replication, then theoretically, there is just one challenge to overcome–to increase the concentration of Zn(II) in virus infected cells.

## 3. Anticoronaviral Properties of Chloroquine and Hydroxychloroquine

### 3.1. Zinc(II) Ionophores and Other Possible Modes of Action

In the context of searching for substances that help to transport Zn(II) ions from the external environment into the cell, all eyes are set on chloroquine and its less toxic analogue, hydroxychloroquine [4,43,44].

Chloroquine, discovered more than 80 years ago, has been used in clinical practice for over 70 years and its patent has expired (in other words “it is cheap”) and has a reasonably safe clinical safety profile, with the side effects of its intake being severe cardiac arrhythmia/heart failure, hepatic disorders, headache, dizziness and gastrointestinal disorders. It also has to be taken into account that chloroquine interacts with many other drugs (azithromycin, cyclosporine, amiodarone, penicillamine, digoxin, cimetidine) and its intake must be strictly monitored in hospitals [45]. Through the years, it has been used to treat malaria and several other diseases, such as rheumatoid arthritis or systemic lupus erythematosis [46]; although some hypothesis exist, its modes of action against these diseases (especially the autoimmune ones) have not been yet sufficiently understood [47,48,49]. It may be correlated with the inhibition effect of phospholipase A2, interaction with lysosomes, inhibition of phagocytosis, inhibition of peroxide synthesis, increase of intracellular pH and reducted stimulation of CD4 lymphocytes. Chloroquine can also interact with porphyrins, acting as an inhibitor of the biocrystallization of hemozoin (a disposal product formed from the digestion of haemoglobin, non-toxic for parasites) [47,48,49].

Now (30.07.2020), with 329 ongoing, promising CQ and/or HCQ anti COVID-19 clinical trials, it is crucial to find out the precise mode of their anti-COVID-19 action in order to be able to design more suitable treatments with less side effects.

Studies performed in 2014 by Xue et al. [2] (we are now referring to the paper which started the avalanche of Zn(II)-ionophore-anti COVID-19 discussions) brought new insight into chloroquine’s activity, reporting significantly elevated intracellular Zn(II) levels when both Zn(II) and chloroquine were added to the cell culture medium, which indicates that chloroquine may act as a Zn(II) ionophore that targets Zn(II) ions into the cell and/or into lysosomes.

Chloroquine and hydroxychloroquine have also been reported to be antiviral agents in papers, where the use of Zn(II) is not mentioned [50,51]. So far, the effectiveness of chloroquine or hydroxychloroquine has been tested against five of seven identified human coronaviruses. They are members of two viral genus: alphacoronaviruses (Human coronavirus HCoV-229E) and betacoronaviruses (SARS-CoV, SARS-CoV-2, MERS-CoV and HCoV-OC43). A simplified classification of the mentioned viruses is presented in Figure 2 [52].

Chloroquine can inhibit SARS-CoV infection at micromolar (IC_50_ = 8.8 µM) concentrations. Interestingly, this IC_50_ is similar to the chloroquine concentration in plasma during malaria treatment [53]. In the suggested mechanism of action, chloroquine (a weak base) causes an increase of endosomal and lysosomal pH, which inactivates the host’s pH-dependent enzymes necessary for viral entrance inside the host cell [51,54,55]. To enter, SARS-CoV interacts with angiotensin-converting enzyme 2 (ACE2) receptor by the spike (S) proteins. Two functional domains of these proteins can be distinguished—S1, responsible for receptor binding and S2—for viral and cellular membrane fusion with specified fusion peptide region, interacting with lipid bilayer [56,57].

After virion uptake, the S protein is specifically cleaved to expose the fusion peptide region. Enhancing the lysosomal pH, chloroquine inactivates pH-dependent lysosomal proteases like cathepsin L necessary for S-protein cleavage, finally preventing the membrane fusion and viral genome release [56,58]. Chloroquine and hydroxychloroquine reduce the glycosylation of the ACE2 receptor and the viral penetration into the cell, due to the alcalisation of endocytic compartments—when not glycosylated, they will have lower affinity towards the viral spike proteins, which probably impedes the virus—receptor binding, resulting in the inhibition of the infection development [55]. Moreover, in in vitro experiments, chloroquine, administered before exposure to the virus, significantly reduces the development of infection, which indicates not only therapeutic but also prophylactic activity. CQ and HCQ caused an immunomodulating effect and it is assumed that the cytokine storm of SARS-CoV-2-infected patients suffering from COVID-19 will be inhibited [55]. Due to the similarity between the SARS-CoV-2 and SARS-CoV viruses, chloroquine’s mechanism of action against them will most likely be similar. In the case of SARS-CoV-2, in vitro assays show that chloroquine and hydroxychloroquine’s antiviral activity is low (it reaches micromolar concentrations) when administered both before and after viral infection (entry and post-entry stage) [43,59]. Furthermore, both agents inhibit the viral transfer from endosomes to lysosomes, which appears necessary for the release of viral genome [43]. Moreover, CQ and HCQ have anti-inflammatory properties, which may alleviate the inflammatory response in in vivo experiments [43].

Clinical trials done in China have shown that chloroquine is effective against SARS-CoV-2-induced infection, alleviating and shortening the disease course with no severe unfavorable effects [3]. Also clinical trials carried out in France on 20 patients (200 mg hydroxychloroquine sulfate, three times per day during 10 days) have shown that 70% of them were virologically cured 6 days after drug inclusion, compared to 12.5% in the control group [4]. Right now, there are 329 ongoing anti-COVID-19 clinical trials (85 on chloroquine and 244 on hydroxychloroquine)–they are commented on in the next section [60].

In vitro studies of chloroquine and hydroxychloroquine on other coronaviruses showed that chloroquine was effective against MERS-CoV at an early step of infection (statistically calculated effective concentration that induces a specific effect in 50% of experimental population under direct/specific conditions, EC_50_ = 3.0 µM) [61]. Human coronaviruses HCoV-OC43, together with HCoV-NL63, HCoV-229E and HCoV-HKU1 induce approximately 15% of common cold infections all over the world [62,63]. In vitro tests show that HCoV-OC43 is also susceptible to chloroquine (EC_50_ = 0.306 µM). In vivo tests, in the murine model, confirmed chloroquine’s effectiveness against HCoV-OC43, when administered at the initial stage of infection [64]. Replication of another human coronavirus, HCoV-229E, was significantly decreased by chloroquine at 10 and 25 µM concentrations. In this study, chloroquine was suspected of inhibiting the activation of p38 mitogen-activated protein kinases (p38 MAPKs) in infected cells [65]. Many viruses are able to use MAPKs to adapt cellular processes to their own purposes, thus inhibiting the activation of p38 MAPK may reduce viral replication [65,66].

In addition to human viruses, chloroquine antiviral activity was also tested against animal coronaviruses, i.e., feline infectious peritonitis virus (FIPV), also known as feline coronavirus (FCoV), which belongs to the alphacoronaviruses and is responsible for inducing the feline infectious peritonitis (FIP)—a fatal disease in domestic and wild cat species [67,68]. Chloroquine decreased viral replication, which was confirmed by both in vitro and in vivo experiments. Due to the anti-inflammatory effect, chloroquine was suggested as a potential therapeutic agent in FIP treatment [68].

### 3.2. Ongoing Clinical Trials

To date (30th July 2020) there are 2834 clinical studies concerning COVID-19 [60]. Chloroquine and hydroxychloroquine (alone or combined with other interventions such as: other drugs, dietary supplements, biological agents, medical devices and others) are considered as treatment in 329 studies (85 on chloroquine and 244 on hydroxychloroquine (where also some CQ ones are included). Chloroquine/chloroquine phosphate/chloroquine diphosphate and/or hydroxychloroquine are most frequently combined with: (i) anti-infective agents which have been already approved and used as: antibiotics (azithromycin, ceftriaxone, piperacillin-tazobactam, ceftaroline, amoxicillin-clavulanate); other antibacterial drugs (moxifloxacin, levofloxacin, carrimycin); drugs used to treat and prevent HIV/AIDS (darunavir/cobicistat, lopinavir/ritonavir and emtricitabine/tenofovirdisoproxil); other antiviral drugs (remdesivir, oseltamivir, interferon-β1a); receptor antagonists (losartan, anakinra, sarilumab) and others (inhibitor of Janus kinase—baricitinib; (ii) antiinflamatory and immunosuppressive agents (hydrocortisone, tocilizumab); (iii) respiratory system agents (expectorants, bromhexine); (iv) anti-arrhythmia/antihypertensive agents (losartan); (v) anticoagulants (gabexate); (vi) anti-allergic agents (ciclesonide); (vii) antineoplastic agents (topoisomerase inhibitors, pembrolizumab, ofloxacin, moxifloxacin, levofloxacin, interferons); (viii) fibrinolytic agents (defibrotide); (ix) renal agents (ofloxacin, levofloxacin); (x) vasoconstrictor agents (giapreza, angiotensin II) and (xi) micronutrients (ascorbic acid, folic acid, ergocalciferols, hydroxocobalamin, iodine, vitamin D, zinc(II)).

Especially interesting (in particular from the point of view of this review) is the last group, where CQ/HCQ is considered as a treatment, together with zinc(II) and other micronutrients. Eighteen of all clinical studies assume that including zinc(II) as a dietary supplement may be essential for achieving effective treatment or prevention of COVID-19. A brief overview of those studies is given in Table 1 below.

Among the 2834 clinical studies concerning COVID-19 (30.07.2020), 743 of them relate to drug interventions (among them are the 329 chloroquine and hydroxychloroquine ones described above) [60]. Others include: biological studies (for example natural killer (NK) cells, recombinant vaccines, mesenchymal stem cells (MSCs) or convalescent anti-SARS-CoV-2 plasma, diagnostic tests (for example breath test, serological tests, lung ultrasound, new QIAstat-Dx fully automatic multiple PCR detection platform, SARS-CoV-2 IgG Antibody Testing Kit), devices (e.g., BIOVITALS, biosensors, SensiumVitals wearable sensor), different questionnaires and others (e.g., SPIN-CHAT Program or telerehabilitation).

Data from ClinicalTrials.gov [60] show that the largest number of drugs tested are included in the category of anti-infective agents (over 16%), but a significant part are also antihypertensive agents (8.3%), antineoplastic agents (7.0%), central nervous system depressants (5.1%), vasodilator agents (4.4%), anti-inflammatory agents (4.3%), analgesics (4.0%), antirheumatic agents (3.8%), micronutrients (3.6%), gastrointestinal agents (3.4%), anticoagulants (3.1%). The remaining, approximately 40% are drugs belonging to a variety of categories, including among others: respiratory system agents, channel blockers, lipid regulating agents, cardiotonic agents, dermatologic agents, neuroprotective agents and others (shown in Figure 3).

As stated above, hydroxychloroquine, chloroquine and chloroquine diphosphate are some of the most frequently studied. Other drugs, which are not dependent on zinc(II) ions, and which appear in numerous clinical studies are comprehensively listed in the Appendix A (Appendix A).

Non-CQ/HCQ and non-Zn(II) involving in vitro studies are now mainly focused on:(i)S protein-ACE2-mediated system and thus viral entry [87,88],(ii)compounds targeting SARS-CoV main protease [89,90],(iii)agents targeting papain-like protease 2 (PLP2) [91],(iv)agents targeting SARS-CoV RdRp system [92,93], and(v)agents targeting SARS-CoV helicase [94].

## 4. Antiviral Properties of Chloroquine and Hydroxychloroquine

Chloroquine and its derivatives have not only an enormous impact on the treatment of coronaviruses [53,55,59,64] (see the previous section), but are also effective against several others viruses in vitro, including: (i) human immunodeficiency virus (HIV) [95,96,97,98,99,100,101], (ii) hepatitis A (HAV) [102], B (HBV) [103], C (HCV) [104] viruses and autoimmune hepatitis [105], (iii) poliovirus [106], (iv) influenza A [107,108], B [109] and H5N1 [110] viruses, (v) Zika virus (ZIKV) [111,112,113], (vi) Ebola virus (EBOV) [114], (vii) Crimean–Congo hemorrhagic fever virus (CCHFV) [115], (viii) rabies virus [116], (ix) Chikungunya virus (CHIKV) [117,118], (x) Dengue virus (DENV) [119,120,121], (xi) Hendra, Nipah viruses [122], and (xii) herpes simplex virus (HSV) [123]. In this section, we briefly discuss the possible “non-Zn(II)-based” modes of action (if known) of chloroquine and hydroxychloroquine.

Among the above mentioned examples, the most comprehensively described antiviral effects of chloroquine and hydroxychloroquine were those against HIV replication, what was tested both in clinical trials [124,125] as well as in in vitro studies. Cell line models [100,126], peripheral blood lymphocytes and monocytes [101], were used to examine chloroquine/hydroxychloroquine inhibition effect on HIV. The mechanism of anti-HIV effect of chloroquine may be associated with several different pathways: (i) inhibition of HIV-1 integrase, (ii) inhibition of Tat(viral early protein)-mediated transactivation, which leads to the decrease of transcription initiation, (iii) reduction of iron concentrations within the cell and thus affecting the reverse transcription [96,127,128,129], (iv) inhibition of HIV-1 replication, which is affected by the post-transcriptional maturation of glycoprotein gp120 [130], fundamental for viral infectivity. Moreover, Savarino et al. showed that at clinically achievable concentrations, chloroquine reveals a broad-spectrum of anti-HIV-1 and HIV-2 activity, which results from the decrease of the incorporation of glucosamine in the envelope glycoproteins from both HIV-1- and HIV-2-infected cells [100]. This glycosylation process may become a new target for antiretroviral therapies.

Chloroquine was also examined in the in vitro treatment against Dengue virus type 2 (DENV-2), which leads to dengue fever [120], showing: (i) a significant reduction of virus production in infected cells, contrary to untreated cells, (ii) reduction of *DENV-2* replication in cells, and (iii) reduction of proinflammatory cytokine expression [120]. Two clinical trials evaluated the efficacy of chloroquine treatment in patients infected with Dengue virus [131,132], showing that chloroquine does not reduce the duration of viraemia and antigenaemia in dengue patients; however, it does reduce fever clearance time and the incidence of dengue haemorrhagic fever [131]. The second clinical trial revealed the reduction of pain symptoms in DENV-infected patients and an improvement of daily fine fettle during treatment, but, on the other hand, did not alter the duration of the disease or the intensity and days of fever [132].

Other antiviral properties of chloroquine are mostly based on in vitro studies and have not been verified in clinical trials or have not revealed the satisfactory therapeutic effect. In hepatitis C virus infection, chloroquine inhibits the intracellular replication of an HCV replicon in a dose-dependent manner and could be used in hopeful combination therapy for HCV–HIV co-infected patients [104]. In case of influenza viruses, chloroquine have shown: (i) inhibitory effect against the replication of human influenza A virus H1N1 and H3N2 (in vitro but not in vivo) [107,108], (ii) the inhibition of influenza B infection by acting on a replicative step preceding primary transcription of the B virus genome by the addition of the drug at the time of infection (the drug had no effect on virus replication when added after 15 min after infection) [109], (iii) the inhibition of autophagy in mouse lung induced by avian influenza AH5N1 virus, while the viral proinflammatory cytokines were not significantly affected [110]. Chloroquine also inhibits Chikungunya alphavirus (CHIKV) infection in vitro, blocking the production of proinflammatory cytokines and the proliferation of monocytes, macrophages, and lymphocytes [117]. The effect of chloroquine on polioviruses is more modest—it can only redirect the processing of input virions without causing viral uncoating [106].

Chloroquine also has an antiviral activity against Zika virus (ZIKV) in different cell modes: Vero cells, human brain microvascular endothelial cells, human neural stem cells, and mouse neurospheres [111]. These studies prove both the reduction of the number of ZIKV-infected cells in vitro and the inhibition of virus production and cell death without a cytotoxic effect [111].

Promising in vitro results were also obtained for the Crimean-Congo hemorrhagic virus (CCHFV) in the presence of two FDA approved molecules: chloroquine and chlorpromazine [115]. The two drugs inhibited CCHFV virus replication even when given up to 6 -24 h after infection.

Chloroquine has also been tested on animal models. Daily treatment of chloroquine to Ebola virus (EBOV) infected mice caused a significant increase of their survival rate (approximately up to 80%–90%) [133]. On the other hand, Dowall et al. [114] have shown that the replication of EBOV virus is indeed inhibited by chloroquine in vitro, but they did not observe any therapeutic effects on EBOV-infected guinea pigs in vivo. Similar, not effective chloroquine treatment in vivo was observed for Nipah and Hendra viruses (paramyxoviruses that cause severe disease such as encephalitis in animals and humans), in golden hamster models [122].

## 5. Other Antiviral Zn(II) Ionophores

Chloroquine and hydroxychloroquine are not the only antiviral zincophores - there are numerous other examples where inhibition of viral replication is observed (mostly in vitro). The replication of various viruses, including: (i) respiratory syndrome coronavirus, SARS-CoV [1], (ii) arteriviruses (equine arteritis virus, EAV [1], (iii) respiratory syncytial virus [26], (iv) porcine reproductive and respiratory syndrome virus, PRRSV [134] (v) picornavirus (poliovirus, Human rhinovirus, Foot-and-mouth disease virus (FMDV)) [27,30,31,32],(vi) herpes simplex virus (HSV) [135] and (vii) influenza virus [25], is inhibited by the simultaneous combination of two approaches: (i) high zinc ion concentration and (ii) the addition of compounds that induce cellular import of Zn(II), such as hinokitol (HK), pyrrolidine dithiocarbamate (PDTC) or pyrithione, (PT) (Figure 4). The molecular mechanism of viral inhibition is most likely based on the rapid import of extracellular Zn(II) ions into the cells by ionophores, after which the imported Zn(II) ions inhibit the RNA polymerase activity and thus, replication [1,32,134].

### 5.1. Pyrithione

For herpes simplex virus (HSV), a more detailed mechanism of inhibition by pyrithione (PT, 1-hydroxypyridine-2-thionine, omadine, Figure 4) was described [135]: (i) PT inhibits HSV late gene expression (e.g., that of glycoprotein D) and viral progene production, in a Zn(II) concentration dependent manner; (ii) PT blocks the HSV immediate early gene expression, and (iii) PT administration may inhibit the cellular ubiquitin-proteasome system (UPS), leading to the interruption of IκB-α degradation and NF-κB activation thus causing the reduction of viral infection symptoms. Pyrithione also shows anticancer, antimicrobial and antifungal properties, which are described in the next sections.

### 5.2. Quercetin

Quercetin (3,3′,4′,5,7-pentahydroxyflavone, Figure 4), found in onions, many Chinese herbs, vegetables, fruits and wines [136], is another zinc(II) ionophore, which shows antiviral effects against following viruses: (i) influenza A virus [137,138], (ii) hepatitis B (HBV) [139] and C (HCV) [140,141] viruses, (iii) Ebola virus (EBOV) [142], (iv) Dengue virus (DENV) [143], (v) Japanese Encephalitis Virus (JEV) [144] and (vi) Zika virus (ZIKV) [145]. The development of an influenza infection is initiated by hemagglutinin (HA), an influenza virus envelope protein, which helps in viral entry into the cell [146]. Mature HA is a homotrimer, and each monomer is consisted of two disulfide-linked subunits, HA1 and HA2 [146]. The mechanism of influenza A virus inhibition by quercetin is strictly related with virus entry into the cell [138]. Quercetin interacts with influenza hemagglutinin protein (most likely with the HA2 subunit) and thus inhibits the formation of the virus-cell link (in vitro studies) [138].

In case of hepatitis viruses, quercetin is responsible for Hepatitis B surface antigen (HBsAg), Hepatitis B antigen (HBeAg) and viral DNA level reduction in vitro. Furthermore, treatment with quercetin also decreases HCV virions integrity, HCV NS3 protease effectiveness and viral transcription levels [139,140,141]. Quercetin has also shown an antiviral efficiency in combination with other compounds. *Houttuynia cordata* Thunb. (HCT) extract, administered together with quercetin, quercetrin and cinanserin mixture showed an inhibitory effect on mouse hepatitis virus (MHV) and Dengue virus type 2 (DENV-2)in vitro [143]. Baicalein and quercetin combination may decrease the amount of the copied RNA of Japanese encephalitis virus [144]. Quercetin derivatives also exhibit effective antiviral activity. Quercetin 3-β-O-D-glucoside was successfully tested against Ebola [142] and Zika [145] viruses, nevertheless, further studies should be performed to elucidate the mechanism of its antiviral action.

### 5.3. Dithiocarbamates

Dithiocarbamates (DTCs, Figure 4), less-known Zn(II) ionophores, exhibit a broad spectrum of antitumor activities [147,148], and may serve as a potential anti-HPV virus drug candidates [149]. Moreover, diethyldithiocarbamate (DDTC), a derivative of DTCs, was also successfully used in patients with HIV-1 infection to delay progression of AIDS in clinical trials [150]. Zincophorin (Figure 4), a polyoxygenated ionophoric antibiotic, first isolated from *Streptomyces griseus* in 1984 [151,152], is active against Gram-positive bacteria and *Clostridium coelchii* in vivo (more details are given in Section 7) [152,153]. Interestingly, its methyl ester, exhibits a potent inhibitory properties against influenza virus with decreased toxicity for the host cells [154,155].

## 6. Anticancer Properties of Zinc(II) Ionophores

Zinc ionophores have a possible role not only in antiviral therapy, but also show anticancer and antibacterial properties. In this section, we discuss the most commonly used and most promising anticancer ionophores, leaving the reader with an open question—could they also be of any use in anti-COVID-19 therapy?

Metal ion homeostasis is crucial for the survival of both normal and cancer cells. Therefore, metal-binding compounds have been considered as potential anticancer agents [156]. Zn(II) ions are involved in cellular signal transduction, DNA synthesis and gene transcription, and are also essential for cell proliferation and differentiation [157]. Increased intracellular zinc concentration stabilizes hypoxia-inducible factor-1 (HIF-1) and therefore has been associated with glycolysis, angiogenesis and apoptotic cell death [158]. For these reasons, Zn(II) homeostasis is targeted to develop novel therapeutic agents for cancer treatments [159]. Two different mechanisms of ionophore-based anticancer activity can be distinguished. Some compounds acts via metal ion chelation. The chelators may bind the metals extracellularly, deplete the metal ions from cancer cells or inhibit molecular pathways, without necessarily removing metal ions [160,161,162]. Other anticancer drugs increase intracellular metal concentrations [159]. Here, we focus on the second mode of action and discuss Zn(II) ionophores.

### 6.1. Pyrithione

Pyrithione is a well-known antimicrobial and antifungal agent [163]. Its anticancer activity was first reported in the 1980s. However, due to its low water solubility (2.5 g/L at 20 °C) and poor bioavailability it is not appropriate for parenteral administration and is used only as a topical agent [164]. Nevertheless, several water-soluble derivatives have been synthesized so far. They increase the pool of free Zn(II) ions and exhibit an anti- proliferative activity in A549 human lung cancer cell lines [158]. The treatment to be effective, it requires Zn(II) supplementation. This approach may be particularly important for prostate cancer - in normal healthy tissues, relatively high levels of Zn(II) ions are found, but this pool decreases drastically upon malignant transformation [165].

### 6.2. Clioquinol

It was shown that also clioquinol (5-chloro-7-iodo-8-hydroxyquinoline, Figure 4) acts as an ionophore and induces apoptosis of human cancer cells [166]. It should be kept in mind that clioquinol, as a metal chelator, possesses antimicrobial activity and is considered as a potential therapeutic in Alzheimer’s disease [167]. Although clioquinol induces inhibition of superoxide dismutase-1 (SOD1), the addition of metals essential for the enzyme’s activity, (which were expected to reverse the activity of the—presumably—sequestering agent) does not diminish cell damage, but on the contrary, enhances cell apoptosis. The obtained results do not prove its metal chelation mode of action hypothesis, and strongly suggest that the addition of clioquinol increases the intracellular zinc pool, suggesting it acts as a zinc ionophore [166].

At this point it is worth to mention, that clioquinol has an interesting combination of properties, namely moderate metal affinity and intermediate p*Ka* value. It allows metal binding outside the cell where a higher concentration of transition metal ions is observed (and also a higher pH), helping them to cross the cell membrane and then releasing metal ions where their concentration is lower (or a decreased pH is observed/competitive ligands are present). It may be especially important in the treatment of cancer—within neoplastic cells, lower pH and increased concentration of glutathione, comparing to healthy cells, is always observed [168]. Therefore, clioquinol has an anticancer effect both in vitro and in vivo (e.g., on DU 145, a human prostate cancer line, and several other human cancer cell lines) [166], however, its mode of action is not really clear. It has been shown that increased concentrations of free zinc(II) occurs in lysosomes, what may suggest that these organelles are primary targets of ionophores. Moreover, the addition of clioquinol and Zn(II) results in the degradation of BH3-interacting domain—death agonist (proapoptotic single domain BH3 proteins which act as apoptotic signal amplifiers or programmed cell death pathway activatiors), which plays a major role in lysosome-mediated apoptotic cell death. Clioquinol treatment results in the increase of free Zn(II) in lysosomes what leads to their disruption, thus apoptotic cell death of cancer cells is induced [169]. Schimmer and coworkers have recently demonstrated that clioquinol inhibits the proteasome through a copper-dependent mechanism and therefore inhibits growth of leukemia and myeloma cells [170]. It was also found that it inhibits NF-κB (nuclear factor κB) activity in ovarian cancer cells [171].

### 6.3. Dithiocarbamates

Another group of metal-binding drugs, which are capable to transport ions across cell membranes, are dithiocarbamates (Figure 4). They are widely used as insecticides, fungicides, therapeutic agents for alcoholism and metal intoxication [172], acting as a metal chelators and SOD1 inhibitors [173,174] or interacting with transcription factors like NF-κB and p53 [175,176,177]. Some dithiocarbamates, such as (the also antiviral and antibacterial) pyrrolidine dithiocarbamate (PDTC, Figure 4), are considered as a copper ionophores that recruit extracellular copper [178], what leads to the oxidation of endogenous thiols or conformational changes of p53 [176]. It was also shown that PDTC causes cerebral endothelial cell death. Transition metal ions, such as copper or zinc, may enhance the cytotoxic effect of PDTC [179]. It is worth noticing that the potency cytotoxic enhancement of zinc is probably greater than that of copper.

## 7. Antibacterial Zn(II) Ionophores

Ionophore antibiotics constitute a heterogeneous group of antimicrobials which are produced by microorganisms, mainly by various species of *Streptomyces* (Gram-positive spore-forming bacteria). Like other ionophores, they transport ions across cell membranes. It leads to the disruption of cell membrane permeability and results in antibacterial effects [180]. Ionophores are active only against Gram-positive bacteria, since Gram-negative bacteria have an outer membrane, acting as a permeability barrier [181].

Widely used antibacterial group 1 and 2 ionophores such as monensin, salinomycin, narasin, lasalocid, maduramicin and laidlomycin transport alkali and/or alkaline-earth metal cations [180]. In all of the listed ionophores, metal ion binding is supported by intramolecular hydrogen bonds formation between the carboxylic and the hydroxyl groups. The hydrophilic cage is formed for the metal, which becomes surrounded by the hydrophobic part of the ionophore [182].

Ionophores transferring cations belonging to group 1 and 2 of the periodic table are not the only ones exhibiting bactericidal activity. Significant interest has been raised to the antibacterial properties of transition metal complexes with ionophores [183]. Among them, Zn(II) ionophores seem to be those of potentially large application. An increase of the Zn(II) pool above several orders of magnitude above physiological level perturbs zinc(II) homeostasis and leads to a cytotoxic effect on prokaryotes at the concentration above 10^−4^ M [184,185]. It may also act as an antifungal agent, but in general, a higher ionophore concentration is needed to inhibit the growth of fungi than for that of bacteria [186].

### 7.1. Pyrithione

One of the most frequently used zinc ionophore is pyrithione—its antiviral and anticancer properties have already been described in the previous sections. This versatile compound has a long history as a safe and effective compound, which has been used for more than 50 years. It was intensely studied in the 1950s for its fungistatic and bacteriostatic properties. It is one of the ingredients of anti-dandruff shampoos due to its antifungal properties against scalp fungus *Malassezia* [164]. It is also widely used in other branches of cosmetology because of its broad antimicrobial activity [187]. Nowadays, Zn(II) pyrithione is also known as a compound which neutralizes *P. aeruginosa, A. baumannii* and *S. aureus*—bacterial species found in human biofilms, also associated with wound infections resistant to conventional antibiotics. Moreover, zinc pyrithione exhibits a synergistic effect with a leading medicament used to treat wound infections, namely silver sulfadiazine. The combination of these compounds, may effectively inhibit disease progression [188].

### 7.2. 8-Hydroxyquinoline

Another important zinc ionophore is 8-hydroxyquinoline (PBT2, Figure 4**)** which often serves as a scaffold or pharmacophore in the design of novel potential drugs with various applications: from antibacterial agents throughout antiviral, antimalarial, antifungal to anticancer and neuroprotecting compounds [189]. It is well known that 8-hydroxyquinoline and its derivatives are excellent metal ion chelators. They form mono or bidentate complexes, in which heterocyclic nitrogen and phenolate oxygen serve as donor atoms [190,191]. It was shown that these compounds act as a zinc and copper ionophores in mammalian cells [192].

Hydroxyquinoline has been developed as a potential drug for Alzheimer’s and Huntington’s disease, however it failed efficacy checkpoints in the phase 2 human clinical trials [193]. Since PBT2 is an ionophore facilitating the transport of first-row transition metal ions across cell membrane, it alters intracellular metal homeostasis and thus exhibits antimicrobial activity. The precise mode of its antimicrobial action remains unclear, but involves at least few mechanisms, which makes the development of PBT2 resistance unlikely. The bactericidal mechanisms of its action include: (i) binding of extracellular zinc, transporting metal ion throughout biological membrane and dissociation within the cytoplasm. A concentration-dependent zinc accumulation with increasing concentrations of PBT2 is observed. Moreover, it was shown that increasing intracellular zinc(II) levels has no effect on the membrane potential, making the whole process electroneutral, (ii) dysregulation of the *mtuABC* expression and manganese depletion and (iii)disturbance of the intracellular redox balance and induction of oxidative stress via accumulation of reactive oxygen species. Moreover, antioxidant capacity of the cell is diminished due to decreased level of manganese, making the cell more susceptible to toxic hydrogen peroxide and superoxide anion radical. As a result of Fenton reaction, hydroxyl radical may be formed, leading to oxidative damage of DNA and causing lethal effect to the bacterial cell [194].

### 7.3. Zincophorin

Many polyoxygenatedionophore-containing natural products exhibit potential antibiotic properties. One of them is zincophorin (Figure 4)—the one whose methyl ester turned out to be an inhibitor of the influenza virus (as mentioned in Section 5). It is a member of polyketide antibiotic family and possesses high activity against Gram-positive bacteria, including *Clostridium perfringens*. Antimicrobial activity of zincophorin has been extended to *Streptococcus pneumonia* lately [195], the leading cause of bacterial pneumonia [196]. In general, minimal inhibitory concentration (MIC) and minimal biofilm inhibitory concentration (MBIC) of zincophorin for three reference strains and four clinical isolates of *S. pneumoniae* are lower than for daptomycin [196] (a relatively new lipopeptide antibiotic used for the treatment of gram-positive life-threatening infections) [197]. It is worth mentioning that zincophorin losses its antimicrobial activity upon esterification of its carboxylate group. This functional group is crucial both for the biological effect (antibacterial and cytotoxic) and for metal ion binding [189]. Zincophorin is able to bind divalent cations, with the stability order of Zn^2+^ ≈ Cd^2+^ > Mg^2+^ > Sr^2+^ ≈ Ba^2+^ ≈ Ca^2+^ [182].

### 7.4. Pyrrolidine Dithiocarbamate

As already mentioned, pyrrolidine dithiocarbamate (PDTC) is a versatile compound with antiviral and antibacterial properties [179]. Due to the presence of two thiol groups in its structure, PDTC has heavy metal-chelating and free radical-scavenging properties. Moreover, PDTC is widely used to inhibit the expression of inflammatory genes [198]. While the coexistence of microorganism infection and inflammatory response are the cause of many diseases, finding the compounds having an appropriate dual mode of action is highly desired [199]. Recent studies revealed that PDTC also exhibits antibacterial activity. It inhibits bacterial growth, e.g., that of *P. gingivalis* (low dose complete response, MIC = 1 mM), *A. actinomycetemcomitans* and *S. aureus* (30-times higher dose, MIC = 30 mM). The antibacterial activity of PDTC is reduced by the addition of copper(II) ions. In contrast, it is enhanced significantly by zinc(II) ions. Obtained results suggest that zinc ions are necessary for antibacterial activity of PDTC, but the precise mechanism of action is not fully elucidated and a more detailed explanation is needed. Nevertheless, PDTC as an anti-inflammatory-antimicrobial-antiviral agent may be useful in the treatment of inflammatory diseases of unknown (bacterial/viral) origin [200].

## 8. Conclusions

Connecting the fact that Zn(II) is an RNA-dependent RNA polymerase inhibitor and the fact that CQ and HCQ are Zn(II) ionophores leads to an avalanche of concepts. We have comprehensively reviewed the latest available information, describing details of the enzyme itself, showing where Zn(II) and CQ/HCQ (taken together, or independently) show antiviral, and in particular–anticoronaviral action. FDA–approved Zn(II) ionophores (or even those which are present in food) with antiviral, but also anticancer and antibacterial properties are intensively discussed. The exponentially growing amount of CQ/HCQ-zinc(II) clinical trials is shown within the framework of other ongoing trials.

So, is Zn(II) the éminence grise? The evidence shown in this work let us think that Zn(II) indeed may play a role in the viral cell cycle, although the mechanism still remains uncertain. The idea of zincophore-mediated Zn(II) uptake is definitely one of the well-defined paths that could lead to the development of new anticoronaviral therapeutics. Although we cannot give a simple one-word answer to the question in the title, we can definitely confirm that it is necessary to continue working on Zn ionophores as potential anti-SARS-CoV-2 agents.

It is important to underline once again that this work by no means serves as a piece of medical advice and does not confirm that taking Zn(II) supplements and eating quercetin-rich raspberries/onions/wine will make anyone more immune to SARS-CoV-2. However, as chemists, we hypothesise that there could be such a possibility and if 24 weeks of clinical trials on CQ/HCQ and Zn(II) [69,70,71] turn out to be a success, this could be a very good direction to go–looking for safer zinc(II) ionophores with less side effects.

## Figures and Tables

**Figure 1 pharmaceuticals-13-00228-f001:**
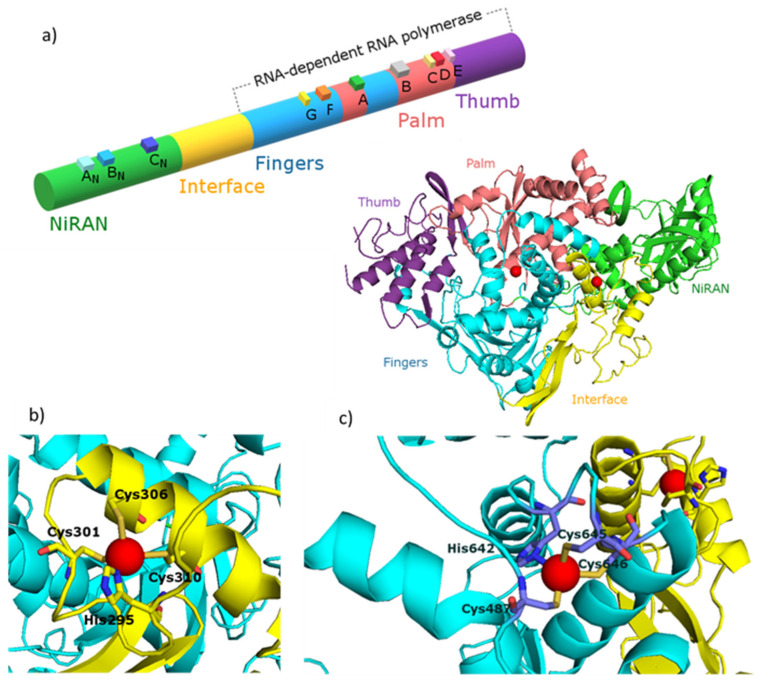
(**a**) Diagram and structure of the *SARS-CoV* nsp12 protein indicating protein domains (NiRAN, interface, fingers, thumb and palm), conserved motifs (A_N_, B_N_, C_N_, G, F, A, B, C, D) and zinc(II) binding sites (red spheres). Enlarged zinc(II) binding sites of nsp12 protein placed in: (**b**) interface region and (**c**) fingers region. The figure was generated using PyMOL [41]. PDB entry: 7BTF [42]. Figures based on [21,22].

**Figure 2 pharmaceuticals-13-00228-f002:**
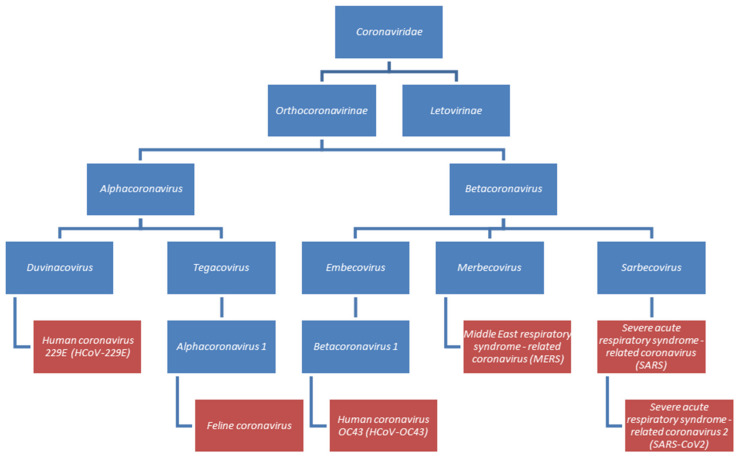
Classification of chosen coronaviruses. Species described in this section were marked in red.

**Figure 3 pharmaceuticals-13-00228-f003:**
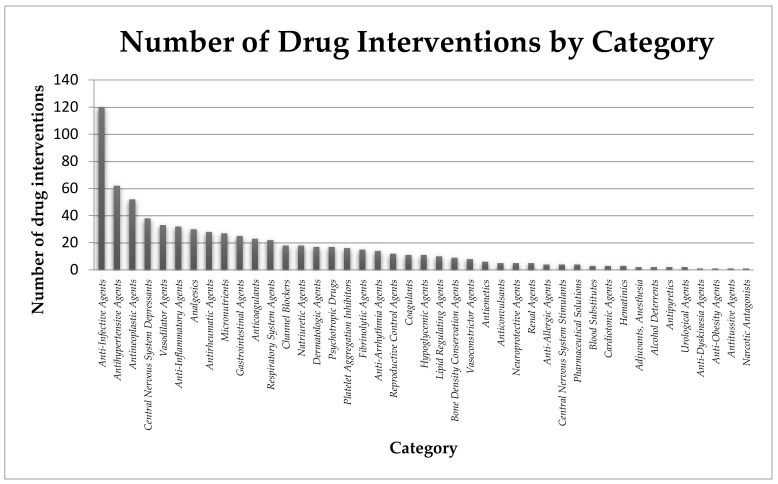
Potential drugs involved in clinical trials against COVID-19 classified in individual categories [60].

**Figure 4 pharmaceuticals-13-00228-f004:**
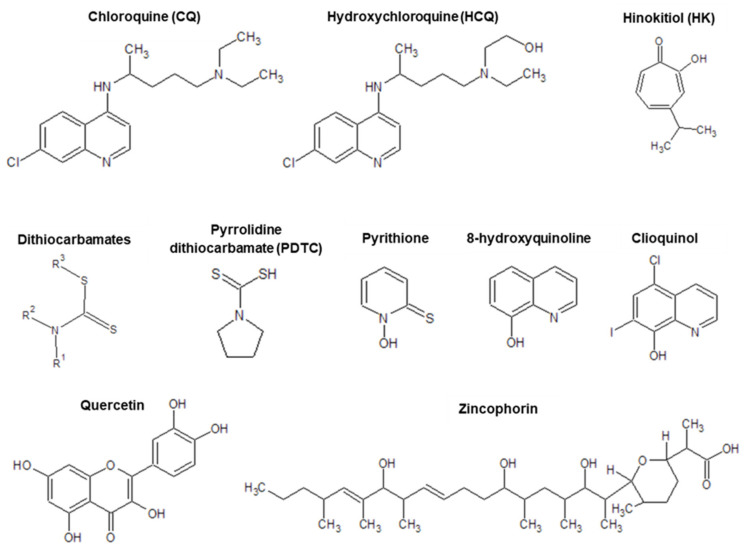
An overview of zinc(II) ionophores discussed in this work: chloroquine (CQ), hydroxychloroquine (HCQ), hinokitol (HK), dithiocarbamates (DTCs), pyrrolidine dithiocarbamate (PDTC), pyrithione (PT), 8-hydroxyquinoline (PBT2), clioquinol, quercetin and zincophorin.

**Table 1 pharmaceuticals-13-00228-t001:** Characteristics of clinical trials concerning zinc(II) as a treatment of COVID-19.

Main Drug	Other Interventions	Clinical Trial Identifier	Duration	Location	Purpose	Estimated Enrollment	Inclusion Criteria:	Ref.
Hydroxychloroquine	Vitamin C, vitamin D, zinc(II)	NCT04335084	24 weeks	Ventura, CA, USA	prevention	600 participants, medical workers who are exposed to COVID-19	Patients >18 years old without COVID-19 symptoms; male or female that are considered to be high-risk individuals	[69]
Hydroxy-chloroquine (Plaquenil) (200 mg)	Vitamin C, vitamin D, zinc(II)	NCT04326725	16 weeks	Istanbul, Turkey	prevention	80 participants, healthcare professionals and their first degree relatives	Patients aged between 20 and 90 years without COVID-19 symptoms; male or female	[70]
Hydroxy-chloroquine (Plaquenil)	Azithromycin (Zithromax), vitamins C and D Zinc(II)	NCT04482686	24 weeks	Ventura, CA, USA	treatment	600 participants, patients with COVID-19 infection	Patients aged between 18 and 75 years; healthy male or female with positive test for COVID-19; diabetic and obese (BMI > 30) patients will be included in the trial but randomization will be stratified	[71]
-	Oral nutrition supplement (ONS): protein (14800 mg), fat (22200 mg), carbohydrate (25000 mg), EPA (1100 mg), DHA (450 mg), GLA (950 mg), vitamins A (2840 IU), C (205 mg), and E (75 IU), selenium (0.018 mg), zinc(II) (5.7 mg)	NCT04323228	12 weeks	King Saud University Saudi Arabia	supportive care	30 participants patients with COVID-19 infection	Patients aged between 18 and 65 years; male or female in stable condition (i.e., not requiringICU admission)	[72]
-	Vitamin C (8000 mg) Zinc(II) Gluconate (50 mg)	NCT04342728	4 weeks	Cleveland, OH, USA	supportive care	520 patients diagnosed with COVID-19	Male or female patients >18 years old with COVID-19 diagnosed in an outpatient setting with any symptom (fever or chills, shortness of breath or difficulty breathing, cough, fatigue, muscle/body aches, headache, new loss of taste, new loss of smell, congestion or runny nose, nausea, vomiting, diarrhea	[73]
Hydroxy-chloroquine (400 mg, 200 mg)/azithromycin (500 mg, 250 mg) hydroxy-chloroquine (400 mg, 200 mg)/doxycycline (200 mg)	Zinc(II) (220 mg)	NCT04370782	8 weeks	Roslyn, NY, USA	treatment	750 participants with COVID-19 infection	Patients 30 years and older; male or female with high initial clinical suspicion by physician based on signs and symptoms (fever, cough, myalgias, fatigue, shortness of breath)	[74]
Hydroxy-chloroquine (400 mg)	Zinc(II) (15 mg)	NCT04377646	8 weeks	Tunis, Tunisia	prevention	660 participants military health professionals exposed to SARS-CoV-2	Patients aged between 18 and 65 years; male or female without COVID-19 symptoms and with negative SARS-CoV-2 diagnostic test; no self-medication with chloroquine, hydroxychloroquine or antivirals	[75]
-	Vitamin D3 (2000 IU) Zinc(II) (30 mg)	NCT04351490	8 weeks	Lille, France	treatment	3140 participants, 60 years and older patients infected with COVID-19	Patients 60 years and older; male or female; institutionalized	[76]
Hydroxy-chloroquine, azithromycin, favipiravir (1800 mg, 800 mg)	Zinc(II)	NCT04373733	4 weeks	London, UK	treatment	450 participants infected with COVID-19	Male or female patients > 18 years old with suspected or confirmed COVID-19 infection (fever ≥37.8 °C and at least one of the following respiratory symptoms, which must be of acute onset: persistent cough, hoarseness, nasal discharge or congestion, shortness of breath, sore throat, wheezing or sneezing)	[77]
Chloroquine	Zinc(II)	NCT04447534	2 weeks	Tanta, Egypt	treatment	200 participants infected with COVID-19	Patients 18 years and older with positive COVID-19 (inclusion/exclusion criteria not very precise; expected treatment outcome difficult to predict)	[78]
Hydroxy-chloroquine (400 mg, 200 mg), azithromycin (500 mg, 250 mg)	Vitamin C (400 mg/kg/day, 200 mg/kg/day Zinc(II) (30 mg) Vitamin D3 (5000 IU) Vitamin B12 (0.5 mg)	NCT04395768	18 months	Melbourne Australia	treatment	200 participants infected with COVID-19	Patients who were > 18 years old; male or female with active COVID-19 infection	[79]
Hydroxy-chloroquine (800 mg, 400 mg)	Zinc(II) (66 mg)	NCT04384458	7 weeks	Fortaleza, Ceará, Brazil	prevention	400 healthcare workers exposed to SARSCoV-2	Patients aged between 18 and 70 years; male or female which works in a healthcare facility delivering direct care to patients with either proven or suspected COVID-19	[80]
Hydroxy-chloroquine (400 mg; 200 mg)	Ivermectin (12 mg) Povidone-Iodine Zinc(II) (80 mg) Vitamin C (500 mg)	NCT04446104	6 weeks	Singapore	prevention	5000 participants, migrant workers at High-risk of COVID-19	Patients aged between 21 and 60 years; male with weight more than 40 kg	[81]
Nitazoxanide, ribavirin, ivermectin	Zinc(II)	NCT04392427	2 years	MansouraEgypt	treatment	100 participants infected with COVID-19	Patients 12 years and older; male or female with the COVID-19 PCR positive test and referred to the quarantine; patient without any comorbidities and no sensitivity or contraindication to the three drugs	[82]
Hydroxy-chloroquine (200 mg)	Azithromycin (500 mg, 250 mg) Zinc(II) (220 mg)	NCT04412746	3 months	Algeria	treatment	100 participants with diabetes and infected with COVID-19	Patients 16 years and older; male or female; hospitalized patients with diabetes known before the admission; new onset diabetes discovered at admission	[83]
Quercetin (500 mg)	Bromelain (500 mg) Zinc(II) (50 mg) Vitamin C (1000 mg)	NCT04468139	4 weeks	Saudi Arabia	treatment	60 participants infected with COVID-19	Male or female patients aged ≥ 18 years with moderate to severe disease (inclusion/exclusion criteria not very precise)	[84]
Ivermectin (0.2 mg/kg)	*Nigella Sativa* (80 mg/kg) Zinc(II) (20 mg)	NCT04472585	4 weeks	Punjab, Pakistan	treatment	40 participants infected with COVID-19	Male of female patients > 18 years old with a positive SARS-CoV-2 PCR test; patients with mild to moderate disease	[85]
Hydroxy-chloroquine	Vitamin C (2000 mg) Vitamin D (Alfacalcidiol 0.001 mg) Zinc(II) (50 mg) Paracetamol	NCT04491994	3 weeks	Punjab, Pakistan	treatment	500 participants infected with COVID-19	Patients aged between 18 and 80 years; male or female, with mild disease	[86]

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
