# Peer review of "Zinc(II)—The Overlooked Éminence Grise of Chloroquine’s Fight against COVID-19?"

_pharmaceuticals, 2020, doi:10.3390/ph13090228_

Round 1
Reviewer 1 Report
An interesting review of zinc and chloroquine as well as other ionophores. Comprehensive. Perhaps a bit overambitious in scope, in that at times it veers off from the topic of zinc and chloroquine or other ionophores. It may be worth removing some of the parts that are off this topic.
Author Response
Dear Reviewer,
Thank you very much for your positive comments.
Indeed, the work veers off the topic of zinc and chloroquine in chapter 2, when we discuss Zn(II) as an RNA-dependent RNA polymerase inhibitor, but this piece of knowledge seems to be crucial for the understanding of how Zn(II) may work.
We also go ‘off topic’ in the supplement, where, in the table, we list the drugs that are now used in ongoing clinical trials. This table, first quite modest, gained an impressive shape at the time we finished writing this paper. That is why we decided to move it to the supplement.
With our best regards,
Authors
Reviewer 2 Report
The manuscript is a well written review focusing on the important general role of zinc in antiviral therapeutic approaches. In combination with CQ and HCQ which act as zinc ionophores, zinc has been shown by several groups to elicit beneficial therapeutic effects, especially when used in the early phase of disease.
The impact of the manuscript would be stronger by adding at least one paragraph which discusses the dosage and the timing of the COVID-19 treatment with HCQ and zinc. In this regard, the recently published preprint.org paper "COVID-19 outpatients - early risk-stratified treatment with zinc plus low dose hydroxychloroquine and azithromycin: A retrospective case series study" should be cited, for example.
In addition, the aspect of timing and dosage would be made clearer for the reader, especially in the light of the misleading information that had been provided by several clinical study reports concluding on the failure of CQ/HCQ, probably due to the late onset of therapy. The authors are encouraged to amend table 1, e.g. by including the "success" and "failure" of individual clinical studies in the light of early/delayed onset of therapy and in the light of dosage and stratification. Stratification is an important tool to exclude patients with co-morbidities such as cardiovascular diseases and unappreciated interactions with other therapeutics.
Author Response
Dear Reviewer,
Thank you very much for your positive comments.
The dosage and the timing of the COVID-19 treatment with HCQ and zinc is indeed of great importance. In Table 1 of the revised manuscript, we included an additional column which discusses this issue. As for the "success" and "failure" of individual clinical studies in the light of therapy onset, dosage and stratification – they are difficult to predict in the listed clinical trials, that have not been completed yet. We comment, where possible, on the expected outcome, however this is only our suggestion based on trial conditions concerning zinc(II)/HQ as a treatment of COVID-19 described at https://clinicaltrials.gov/.
We also included a paragraph on the dosage and timing of Zn(II)/HQ treatment (thank you for giving us this idea):
‘Another work by Scholz et al. [doi: 10.20944/preprints202007.0025.v1] describes outcomes of the treatment of COVID19 patients after early treatment with zinc(II), low dose hydroxychloroquine, and azithromycin (the triple therapy) dependent on risk stratification. The study, which includes a treatment group of 141 COVID19 patients (median age 58 years) treated with the triple therapy for 5 days, shows that the triple therapy treatment of COVID-19 should be included as early as possible after symptom onset. The zinc(II) and hydroxychloroquine treated group was associated with significantly less hospitalizations (84% less than in the untreated control group) and 5 times less all-cause deaths (one patient (0.7%) died in the treatment group versus 13 patients (3.5%) in the untreated group). The study further underlines the importance of the early onset of Zn(II) treatment, which makes perfect sense if we consider Zn(II) as an RNA-dependent RNA polymerase inhibitor (as discussed in chapter 2).’
With our best regards,
Authors
Reviewer 3 Report
Zinc + chloroquine/hydroxychloroquine (CQ/HCQ) in the treatment of hospitalized COVID-19 patients may help to improve clinical outcomes and to limit the COVID-19 fatality rates. Therefore, whether zinc supplementation in combination with CQ/HCQ should be recommended for high risk or also younger patients outside of clinical trials as a prevention or treatment approach during SARS-CoV-2 pandemic, should be considered only on a case-by-case basis/
I think that this manuscript from above-abstract may be recognized to be very appreciable for your Journal Publication, please accept this article on your Journal publication.
Author Response
Dear Reviewer,
Thank you very much for your positive comments. We do hope that outcomes of further clinical trials will remain optimistic.
With our best regards,
Authors